# Emergency Department Admissions of Children with Chest Pain before and during COVID-19 Pandemic

**DOI:** 10.3390/children10020246

**Published:** 2023-01-30

**Authors:** Riccardo Lubrano, Vanessa Martucci, Alessia Marcellino, Mariateresa Sanseviero, Alessandro Sinceri, Alessia Testa, Beatrice Frasacco, Pietro Gizzone, Emanuela Del Giudice, Flavia Ventriglia, Silvia Bloise

**Affiliations:** Department of Paediatrics, Sapienza University of Rome, Viale del Policlinico 155, 00161 Roma, Italy

**Keywords:** children, chest pain, emergency, COVID-19, causes, emotional stress, psychogenic origin

## Abstract

Objectives: We compared the number of accesses, causes, and instrumental evaluations of chest pain in children between the pre-COVID-19 era and the COVID-19 period and analyzed the assessment performed in children with chest pain, highlighting unnecessary examinations. Methods: We enrolled children with chest pain admitted to our emergency department between January 2019 and May 2021. We collected demographic and clinical characteristics and findings on physical examinations, laboratory tests, and diagnostic evaluations. Then, we compared the number of accesses, causes, and instrumental assessments of chest pain between the pre-COVID-19 era and the COVID-19 era. Results: A total of 111 patients enrolled (mean age: 119.8 ± 40.48 months; 62 males). The most frequent cause of chest pain was idiopathic (58.55%); we showed a cardiac origin in 4.5% of the cases. Troponin determination was performed in 107 patients, and the value was high only in one case; chest X-rays in 55 cases and echocardiograms in 25 cases showed pathological findings, respectively, in 10 and 5 cases. Chest pain accesses increased during the COVID-19 era (*p* < 0.0001), with no differences in the causes of chest pain between the two periods. Conclusions: The increase in accesses for chest pain during the COVID-19 pandemic confirms that this symptom generates anxiety among parents. Furthermore, our findings demonstrate that the evaluation of chest pain is still extensive, and new chest pain assessment protocols in the pediatric age group are needed.

## 1. Introduction

Chest pain in children is a common cause of pediatric emergency department access [1,2,3,4], representing about 0.3–0.6% of all visits [5]. Considering the association among chest pain, heart disease, and sudden death in adults, the presence of this symptom in the pediatric age group causes concern and alarm in parents with frequent recourse being an evaluation in the emergency department [6].

Conversely, the most common causes of chest pain in children are of non-cardiac origin and include musculoskeletal, idiopathic, and psychogenic origins. In contrast, cardiac causes are identified in a small percentage of cases, ranging from 0% to 10% [7,8,9,10,11,12,13]. 

The diagnostic pathway should include a careful evaluation of the medical history, a physical examination, an electrocardiogram, and eventually targeted testing (such as laboratory tests with cardiac enzymes, chest X-rays, and echocardiograms) based on past medical and family history, physical examinations, and electrocardiographic abnormalities [8,14,15,16,17]. 

Although there has been a growing interest in developing standard guidelines for the management of chest pain in the pediatric age group, to date, the evaluation of chest pain in children follows very different criteria from center to center, with extreme variability in clinical pathways and with a high recurrence of unindicated examinations [17,18,19,20,21]. 

Collins et al. proposed a structured approach to the assessment of chest pain in a child based on a thorough initial consultation (detailed history and physical examination) that allowed to exclude rare and serious diseases and provide vital reassurance to children and families. According to the authors, further investigations and interventions are reserved for those cases where the history and examination do not suggest a diagnosis or concerning features have been identified (for example, acute onset of severe pain, awakening from sleep, history of drooling, foreign body ingestion, cough, fever, dyspnea, history/signs of significant trauma, abnormal pulmonary/cardiac auscultation, ECG for cardiac-type chest pain, etc.) [18]. Other authors demonstrated that by applying the red flags (for example, chest pain with exertion, exertional syncope, past medical history, positive for hypercoagulable state, arthritis, vasculitis, pathological findings on physical examination, or positive family history), it is possible to identify children with chest pain who are in need of a further diagnostic investigation as opposed to pediatric patients with chest pain without red flags who do not have cardiac disease as an explanation for their symptoms. According to the authors, patients with a low risk can be managed successfully by their pediatric provider [19]. Instead, Chen et al. recommend that clinicians evaluate the potential cause of chest pain based on the history of the present illness, past medical history, family history, and physical examination and refer patients to an appropriate specialist. Patients with potential cardiac chest pain should undergo an ECG or ECG plus echocardiogram if they have a pathological heart murmur or hypoxemia, respectively. Patients should undergo myocardial enzyme testing if they have an abnormal ECG plus a normal echocardiogram or normal/abnormal ECG plus abnormal echocardiogram. Patients with non-cardiac chest pain may continue to receive treatment or follow-up if the chest pain alleviates or resolves after treatment, or they should be referred to a pediatric cardiologist for evaluation to exclude heart conditions if the chest pain does not improve after treatment [20]. Finally, Fisher et al. proposed an algorithm for the management of chest pain in children. The authors recommend ECG, IV access, troponine, D-dimero, chest X-ray, and POCUS for ill-appearing patients (abnormal vital signs, altered mental status, abnormal color, perfusion, and hypoxia) or high-risk associated symptoms (pain that is acute associated with exercise or syncope, recent chest surgery, and pain radiation) [21]. 

Even if the impact of COVID-19 on the pediatric population has been mild [22,23], the pandemic has caused a new scenario in pediatric emergency departments characterized by a drastic decrease in accesses and admissions [24,25], with a more significant impact than adult emergency department visits [26].There was an increase in yellow codes and a decrease in green codes, suggesting that parents may have avoided necessary care for their children [27]. Furthermore, there was a change in the main reasons for accesses to pediatric emergency departments, with a higher decrease in air-communicable diseases compared to non-air-communicable diseases [28], mainly related to the implementation of preventive measures also in the pediatric age group [29,30,31]. 

In this context, we wanted to analyze the differences in the causes, number of accesses, and diagnostic evaluations of chest pain in children between the pre-COVID-19 era and the COVID-19 period.

The secondary aim of this study was to analyze the clinical and instrumental assessments performed in children with chest pain, according to our center’s protocol, to highlight any unnecessary and expensive tests performed.

## 2. Materials and Methods

This was a retrospective cohort study performed at our Emergency Pediatric Department (Pediatric Unit of Santa Maria Goretti Hospital, Latina—Sapienza University of Rome). The study was approved by the Institutional Review Board of the Maternal and Child Health Department of the Local Health Authority of Latina (protocol 03-15/03/2021). 

We collected the data of children with chest pain, younger than 16 years, admitted to our Emergency Pediatric Department (ED) during two periods: from January 2019 to 29 February 2020 (pre-COVID-19 era) and from March 2020 to May 2021 (COVID-19 era). We managed the patients according to the protocol of our center (Figure 1).

For each patient, we collected the demographic and clinical characteristics, past medical history, family history (sudden or unexplained death, cardiomyopathy, and severe hyperlipidemia in a first-degree relative), and findings on physical examination and laboratory and instrumental evaluations. 

In detail, we performed physical examinations, laboratory tests, and electrocardiograms on all patients with chest pain. Then, in the case of red flags relating to medical history, pathological findings on physical examination, alterations in cardiac enzymes, or electrocardiographic alterations, the patients underwent an instrumental evaluation with echocardiogram and/or chest X-ray ± lung ultrasound and, if necessary, second-level examinations, such as Chest-Computed Tomography (CT) or Nuclear Magnetic Resonance (RMN). Furthermore, the presence of a SARS-CoV-2 infection was tested in all patients with chest pain since the onset of the pandemic.

*Clinical characteristics* detected of chest pain included onset, frequency, duration, localization, type, association with effort, changes with position or with breathing, and associated symptoms (such as respiratory or gastrointestinal symptoms). 

*Pathological findings* researched on physical examination included acupressure pain at the sternocostal and costochondral joints, heart murmurs, gallop rhythm, pericardial rubs, second-tone doubling, hepatomegaly, reduced peripheral and/or central pulses, peripheral edema, tachypnea, hypertension, tachycardia, fever, crackles, wheezes, or decreased vesicular murmur. 

*The laboratory and instrumental evaluations* included blood count, aspartate transaminase (AST), alanine transaminase (ALT), blood glucose, creatine phosphokinase (CPK), lactate dehydrogenase (LDH), creatinine, c-reactive protein (CRP) and cardiac enzymes (troponin and CK-MB), electrocardiogram (ECG), echocardiogram, chest X-ray, lung ultrasound, and Chest- Computed tomography (CT). 

The echocardiogram was performed by pediatric cardiologist using a My Lab 70 X Vision system (Esaote SpA), equipped with 2–4-MHz transducer. 

The lung ultrasound was performed using a Samsung RS80A US scanner with Prestige equipment (Samsung Medison Co., Ltd., Seoul, Republic of Korea), with a 12 MHz linear transducer. 

The Chest-Computed tomography scan was performed with the latest generation 3D CT and evaluated by our radiologists. 

After medical evaluation with any associated tests, the origin of chest pain was classified as idiopathic, musculoskeletal, respiratory, gastrointestinal, cardiac, psychogenic, traumatic, or other causes. 

Musculoskeletal etiology was diagnosed if the patient presented with pain and tenderness in costochondral junctions, increased pain with muscle strains, and changes in movement. Respiratory cause was diagnosed if the patient presented with respiratory symptoms, such as cough, asthmatic crisis, or radiographic and/or ultrasound signs of bronchitis and pneumonia. 

Gastrointestinal etiology was diagnosed in the presence of chest pain associated with meals, in the presence of heartburn, regurgitation, globus sensation, dysphagia, belching, and vomiting. If necessary, both groups were subsequently referred to pulmonary and gastroenterological specialists. 

Cardiac etiology was diagnosed in the presence of chest pain and electrocardiographic or echocardiographic signs of heart disease.

Psychologically etiology was diagnosed based on of the patient’s history, focusing on the social aspect, school activity, and family life. Patients in turn were referred to a specialist child neuropsychiatrist, if necessary. Traumatic pain was diagnosed based on the history, in particular previous chest trauma, and by chest X-ray and/or ultrasound for possible fractures. 

After careful clinical evaluation, patients without a cause of chest pain, were considered to have pain of an idiopathic nature.

Then, we compared the periods from January 2019 to February 29 2020 (pre-COVID-19 era) and from March 2020 to May 2021 (COVID-19 era). The variables considered were number of accesses, causes and clinical characteristics of chest pain in the two periods. 

### Statistical Analysis

The statistical analysis was performed with JMP® 14.3.0 program for Mac (SAS Institute Inc., Cary, NC, USA). The qualitative variables were described as the distribution of absolute frequencies and percentages, while the continuous variables were expressed as mean ± SD. We used χ^2^ test to compare categorical variables. A *p* value < 0.05 was considered significant.

## 3. Results

Between January 2019 and May 2021, 11,855 patients presented to our emergency department; of these, 111 patients were accessing for chest pain, representing 0.94% of the total emergency accesses of the study period. Specifically, total accesses in the emergency department were 9166 in the pre-COVID-19 era and 2689 in COVID-19 era. 

The mean age was 119. 8 ± 40.48 months; 62 were males (55.9%). None of the patients had a positive past family history of heart disease; 36 patients (32.43%) presented comorbidities, including: gastroesophageal reflux (11.7%), overweight (5.4%), psychiatric disorders (5.4%), asthma (3.6%), renal diseases (2.70%), anemia (1.80%), coronary malformations (1.8%), adenoid hypertrophy (0.90%), and neurofibromatosis (0.90%). 


*Characteristics of chest pain:*
**Localization**: retrosternal area (59.46%) and left hemithorax (25.24%);**Type**: indefinite (86.50%), puncture (7.20%), and oppressive (6.30%);**Pattern**: sporadic (78.38%) and recurrent (21.62%);**Onset**: at rest (60.37%) and during physical exertion (16.21%).


Patient demographic and clinical data are shown in Table 1. 

### 3.1. Comparison between Pre-COVID-19 Era and COVID-19 Era

#### 3.1.1. Number of Accesses

In the pre-COVID-19 era, the number of accesses for chest pain was 68 patients out of a total number of accesses of 9166 (0.74%), while in the COVID-19 era, it was 43 patients out of a total number of accesses of 2689 (1.6%). Therefore, the incidence of chest pain in the pre-COVID-19 period was lower than in the COVID-19 period (*p* < 0.001). 

#### 3.1.2. Causes of Chest Pain 

Regarding the etiology of chest pain, we showed no differences between the two periods: idiopathic causes (pre-COVID-19 period 42 (61.76%) vs. COVID-19 period 22 (51.16%), *p* = 0.20); psychogenic causes (pre-COVID-19 period 5 (7.35%) vs. COVID-19 period 6 (13.95%), *p* = 0.25); gastrointestinal causes (pre-COVID-19 period 5 (7.35%) vs. COVID-19 period 4 (9.3%), *p* = 0.71); respiratory causes (pre-COVID-19 period 6 ( 8.82%) vs. COVID-19 period 2 (4.65%), *p* = 0.40); musculoskeletal causes (pre-COVID-19 period 3 (4.41%) vs. COVID-19 period 3 (6.98%), *p* = 0.3); cardiac causes (pre-COVID-19 period 3 (4.41%) vs. COVID-19 period 2 (4.65%), *p* = 0.95); and traumatic causes (pre-COVID-19 period 3 (4.41%) vs. COVID-19 period 4 (9.3%), *p* = 0.31). 

#### 3.1.3. Laboratory and Electrocardiogram Evaluations 

According to our protocol, all patients performed laboratory tests and electrocardiograms. Laboratory tests were performed in 96.3% of the cases (not in four cases due to the impossibility of blood sampling). Laboratory abnormalities in the pre-COVID-19 period were CPK elevation (eight cases), related to intense sports activity in the previous days and blood glucose elevation (one case) in a patient with diabetic ketoacidosis; laboratory abnormalities in the COVID-19 period were CPK elevation (four cases) related to intense sports activity in the previous days and troponin elevation (one case) related to myocarditis. 

Electrocardiograms were performed in 100% of cases. ECG alterations in the pre-COVID-19 period were pathological ST segment in two cases of pericarditis; there were no ECG alterations in the COVID-19 period. 

#### 3.1.4. Instrumental Evaluation 

Regarding the diagnostic tests, there was the same percentage of use in both periods. 

Chest X-rays (pre-COVID-19 period: 29 (42.6%); COVID-19 period: 26 (60.4%), *p* = 0.61). Of the 29 chest X-rays performed during the pre-COVID-19 period, only 6 were positive showing bronchitis (five cases), pneumonia (one case), while of the 26 chest X-rays performed during the COVID-19 period, 4 were positive, showing bronchitis (one case), COVID-19 pneumonia (one case ), rib fractures (one case), and parenchymal calcifications with pleural effusion (one case). 

Echocardiograms (pre-COVID-19 period: 15 (22%); COVID-19 period: 10 (23.2%), *p* = 0.51). Of the 15 echocardiograms performed during pre-COVID-19 period, only 2 were positive showing two cases of pericarditis, while of the 10 echocardiograms performed during the COVID-19 period, 3 were positive, showing myocarditis (one case) and abnormal origin of the right coronary artery known in history (two cases). 

Chest-computed tomography was performed in two cases, both in the COVID-19 period, following suspicious radiographic findings: sequelae of COVID-19 pneumonia was diagnosed in one case; in the other case, there were micronodules in the right lung with pleural effusion with suspected neoplastic origin. 

Lung ultrasound was performed in one case during the pre-COVID-19 period, confirming pneumonia and in three cases during the COVID-19 period, diagnosing pneumonia, rib fractures, and pleural effusion.

**Table 1 children-10-00246-t001:** Patient demographics and clinical data.

Patients with Chest Pain (n = 111)	
Male, n (%)	62 (55.9%)
Age (mean ± SD), months	119.8 ± 40.48
Positive past family history	0 (0%)
Comorbidities, n (%)	Gastroesophageal reflux: 13 (11.7%) Overweight: 6 (5.4%) Psychiatric disorders: 6 (5.4%) Asthma: 4 (3.6%) Renal diseases: 3 (2.7%) Anemia: 2 (1.8%) Adenoid hypertrophy: 1 (0.9%) Neurofibromatosis: 1 (0.9%) Coronary malformations: 2 (1.8%)
Localizations, n (%)	Retrosternal area: 66 (59.46%). Left hemithorax: 28 (25.24%).
Types, n (%)	Indefinite: 96 (86.50%) Puncture: 8 (7.20%) Oppressive: 7 (6.30%)
Onset, n (%)	At rest: 67 (60.37%) Physical exertion: 15 (13.21%)

## 4. Discussion

Our findings show that there was an increase in emergency department admissions for chest pain during the COVID-19 pandemic compared with the incidence of this symptom during the pre-COVID-19 period.

In a context dominated by fear of the hospital because of the risk of infection, the increase in accesses for chest pain confirms that this symptom, although benign in most pediatric cases, generates anxiety and alarm among parents [32,33,34]. 

This higher incidence in the COVID-19 period probably is due to the emotional stress the children were subjected to related to school closures, lack of outdoor activities, and total disruption of social life. Several studies have reported the limited opportunities for healthy movement behaviors for children and changes in physical activity, screen time, and sleep duration with negative consequences on their health and development [35,36,37]. This likely affected their mental health by generating anxiety, depression, and the appearance of functional symptoms, including chest and abdominal pain [38,39,40]. 

Therefore, we believe it is crucial to ensure that children return to an everyday life, attending school and other extracurricular activities to secure their healthy mental and physical, and relational development; indeed, this needs to be handled safely through the implementation of prevention measures also in the pediatric age group [29,30].

Regarding the etiology of this symptom, we showed no differences between the two periods. Our results confirm that chest pain in children is often a benign condition and cardiac disease remains an uncommon cause in the pediatric age group. In particular, in our study, the cardiac cause was involved in 4.5% of cases. These data are in line with those already present in the literature; in fact, the incidence of a cardiac origin underlying chest pain in the pediatric age group was found in a percentage between 0.6 and 9.27% [41,42,43], with myocarditis and pericarditis identified as the most frequent causes of chest pain in the pediatric age group, as in our cohort of study [44,45,46,47]. 

These causes are not to be underestimated, especially after the cases of myocarditis and pericarditis reported after COVID-19 vaccination in adolescents. However, in our cohort of patients, no cases were related to the SARS-CoV-2 vaccine [48]. 

Furthermore, in agreement with previous studies, among non-cardiac causes, idiopathic pain was the most frequent cause. [2,10,49,50,51,52,53]. 

Not surprisingly, in most cases in our cohort of patients, the pain did not present high-risk characteristics; in fact, it was more frequently of an undefined type, with onset at rest, not linked to changes in position or other associated symptoms.

An interesting aspect that emerged from our study is that despite the frequent benign nature of pediatric chest pain, the use of extensive diagnostic tests is every day in these patients, with an increase in resource use and charges.

We showed that of the 25 echocardiographs performed, only 5 were positive for cardiac etiologies underlying chest pain; similarly, of the 55 chest X-rays requested, only in 10 cases were pathological abnormalities identified. 

Analyzing the cases of patients undergoing echocardiography or chest X-ray, we found that in the patients presenting with pathological findings, the red flags were already present in the patient’s clinical history or physical examination or electrocardiogram. 

In particular, of the five patients presenting with pathological echocardiographic abnormalities, two had a past history positive for an anomaly of coronary origin, two patients with pericarditis presented oppressive and intense chest pain and electrocardiographic abnormalities (pathological ST segment and alterations in the phase of repolarization), and, finally, in one patient with myocarditis, the pain was intense, unresponsive to drugs, and especially associated with palpitations and autonomic symptoms (nausea and loss of appetite). In addition, in the patients presenting with pathological chest X-ray abnormalities, the history and physical examination already identified one suspect of respiratory and traumatic origins of chest pain.

These findings agree with other studies [8,14,15,16], which developed a specific algorithm, named the Standardized Clinical Assessment and Management Plan (SCAMP), based on patient history, physical examination, and electrocardiogram to identify the patients for additional testing. 

In addition, the evaluation of troponin was altered in only one patient, showing that although this examination is easy and inexpensive, its determination should only be reserved for those patients with clinical signs suggestive of heart disease or electrocardiographic abnormalities.

The troponin assay in pediatrics is not well validated. In cases of chest pain in an afebrile patient without electrocardiogram changes, the troponin value is negative in more than 99% of cases [54,55,56]; instead, in the case of fever or ECG graphic alterations, the troponin assay could have a definite diagnostic value for myocarditis and pericarditis, but the prognostic value is low. The diagnosis of our only patient with an increase in troponin was myocarditis, but we want to underline that the patient presented clinical warning signs already on clinical evaluation with an indication to perform an echocardiographic examination, even before the results of the laboratory tests.

Therefore, we believe it is necessary to define new chest pain assessment protocols to select the patients who undergo second-level examinations. In particular, to date, thanks to advances in technology together with increasing accuracy and computational power, we have new and minimally invasive methods, such as point-of-care ultrasound (POCUS), that allow us to diagnose the main respiratory and cardiology conditions quickly, with low cost and without the use of ionizing radiation [57,58,59,60,61,62]. In addition, in our study, pulmonary ultrasonography confirmed the pathological findings shown on radiography, diagnosing three cases of pneumonia and one case of child abuse rib fracture [63].

This method could be used as a first-line diagnostic tool in children with chest pain that based on clinical history, physical examination, and electrocardiographic abnormalities need a more detailed diagnostic investigation. This approach could be advantageous in the evaluation of chest pain in pediatric emergency departments, stratifying patients who should undergo a more detailed evaluation and reducing the use of unnecessary and more costly examinations [64,65]. 

Surely, further studies and multicenter evaluations are needed to modify the diagnostic algorithms of children with chest pain accessing the pediatric emergency department. 

## 5. Conclusions

Our study confirms that pediatric chest pain has a benign origin in most cases. However, the increase in accesses for chest pain, without differences in etiology, during the COVID-19 pandemic confirms that this symptom causes fear and alarm in parents. 

It will be interesting to analyze the effects of returning to normal life for the pediatric age group on the incidence of chest pain accesses in emergency departments

Finally, we showed that the assessment of chest pain in children is still extensive, with frequent recourse to laboratory and instrumental exams unnecessary. 

It is necessary to define new chest pain assessment protocols, inserting the point-of-care ultrasound in the diagnostic pathway, contributing in this way to a better use of economic resources of the health system and improving high-quality care.

## Figures and Tables

**Figure 1 children-10-00246-f001:**
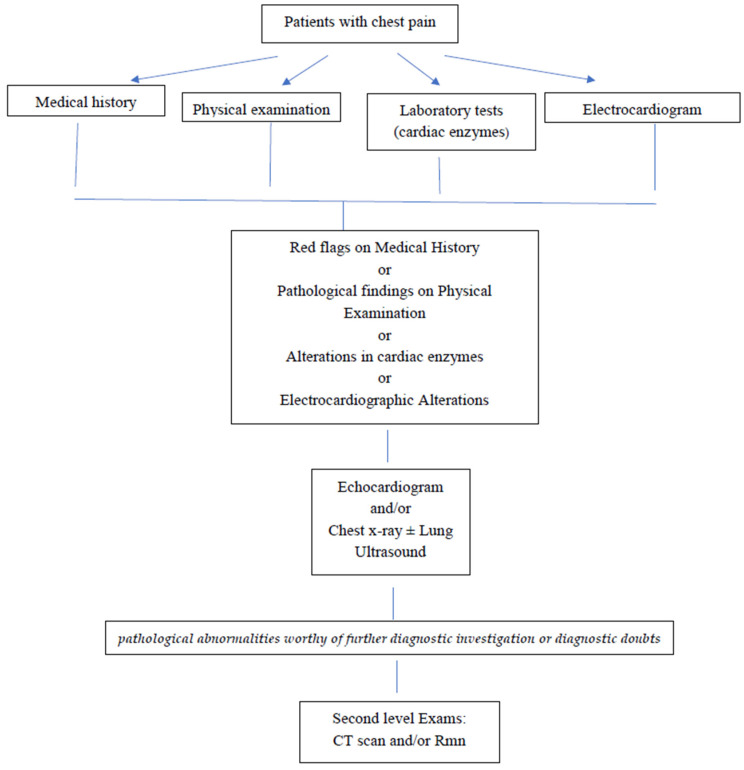
Flow chart for chest pain management in our center.

## Data Availability

All data and materials support published claims and comply with field standards.

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
