# Peer review of "Emergency Department Admissions of Children with Chest Pain before and during COVID-19 Pandemic"

_children, 2023, doi:10.3390/children10020246_

Round 1

Reviewer 1 Report

Dear Authors,

Good study and observations. Please see the suggestion below:Line 9 heading (objectives) needs to be in bold.

Line 22 has word 'that' twice.

Results from line 129 to 139, are either needed to be in bullet format or table format or just keep it simple as paragraph.

paragraph mention causes of pain is redundant since all the values have been put in table format.

Background could be expanded.

Major English check required.

Thanks.

Author Response

Dear Editors and Reviewers,

Thank you for your consideration of our manuscript. We truly think the manuscript is improved after the revisions suggested. Below we respond in detail to the comments and points the reviewer raised. We now submit our revised manuscript for publication in Children. We marked the revisions through the manuscript using the “track changes”.

Reviewer 1

Dear Authors,

Good study and observations. Please see the suggestion below:

Line 9 heading (objectives) needs to be in bold.

We have now modified “objectives” in bold.

Line 22 has word 'that' twice.

We have now deleted the repetition

Results from line 129 to 139, are either needed to be in bullet format or table format or just keep it simple as paragraph.

We have now modified the results from line 129-139 in bullet format and expressed also these results in table 1.

paragraph mention causes of pain is redundant since all the values have been put in table format.

We have now deleted the paragraph mention causes of pain.

Background could be expanded.

We have now expanded the background, adding more references about the evaluation of chest pain in children and about the pediatric care during pandemic.

Major English check required.

We have now revised English language to improve fluency and quality of the paper and corrected spelling and grammatical errors throughout the text.

Thanks.

Reviewer 2 Report

1.       The idea is interesting to compare pre-covid and post-covid children with chest pain presenting to the ER. Is there any reason the patient’s younger than 16 years for the children?

2.       We understand that children are a vulnerable population. Although this retrospective study has been approved by IRB (protocol 03-15/03/2021), the collected information is later (to May 2021), indicating that the study period may not be compatible with the approved period. It is better to provide the approved document in the English version.

3.       In your study, chest computed tomography has been described in the materials and methods. Do all children receive chest CT for the evaluation of chest pain? If yes, why do the children need CT imaging studies to evaluate chest pain? The dose of radiation may be harmful to the children.

4.       The capital letter should be consistent in table 1.

Author Response

Dear Editors and Reviewers,

Thank you for your consideration of our manuscript. We truly think the manuscript is improved after the revisions suggested. Below we respond in detail to the comments and points the reviewer raised. We now submit our revised manuscript for publication in Children. We marked the revisions through the manuscript using the “track changes”.

Reviewer 2

  1. The idea is interesting to compare pre-covid and post-covid children with chest pain presenting to the ER. Is there any reason the patient’s younger than 16 years for the children?

Yes, our Emergency Pediatric Department (Pediatric Unit of Santa Maria Goretti Hospital, Latina - Sapienza University of Rome) provides care up to the age of 16.

  1. We understand that children are a vulnerable population. Although this retrospective study has been approved by IRB (protocol 03-15/03/2021), the collected information is later (to May 2021), indicating that the study period may not be compatible with the approved period. It is better to provide the approved document in the English version.

Yes, we report here the English version of the approved document

On 03/15/2021, we approve the study Emergency room admission of children with chest pain before and during pandemic COVID-19 with data collection from January 2019 to February 29, 2020 (pre-COVID-19 era) and March 2020 to March 2021 (COVID-19 era). If the data collected during the study period are insufficient, the Institutional Review Board allowed the researchers to extend data collection for another 6 months.”

  1. In your study, chest computed tomography has been described in the materials and methods. Do all children receive chest CT for the evaluation of chest pain? If yes, why do the children need CT imaging studies to evaluate chest pain? The dose of radiation may be harmful to the children.

No, chest CT is performed only in the presence of pathological abnormalities worthy of further diagnostic investigation or diagnostic doubts. In fact, in our cohort of patients, chest CT was performed in only 2 cases, one in a girl with previous SARS-CoV-2 infection who presented with chest pain and abnormalities on chest X-ray, one in case of patient with suspected mass of neoplastic origin on chest X-ray. We have modified the flowchart to make our diagnostic algorithm for pediatric chest pain clearer to readers.

  1. 4.       The capital letter should be consistent in table 1.

Yes, we have now corrected this error.

Round 2

Reviewer 1 Report

Dear Authors,

Thanks, for addressing the concerns and suggestions.

Reviewer 2 Report

The authors have replied to the concerns appropriately.